# Photo-Realistic Interactive Virtual Environments for Neurorehabilitation in Mild Cognitive Impairment (NeuroVRehab.PT): A Participatory Design and Proof-of-Concept Study

**DOI:** 10.3390/jcm9123821

**Published:** 2020-11-26

**Authors:** Filipa Ferreira-Brito, Sérgio Alves, Osvaldo Santos, Tiago Guerreiro, Cátia Caneiras, Luís Carriço, Ana Verdelho

**Affiliations:** 1Instituto de Saúde Ambiental (ISAMB), Faculdade de Medicina, Universidade de Lisboa, 1649-028 Lisboa, Portugal; osantos@medicina.ulisboa.pt (O.S.); averdelho@medicina.ulisboa.pt (A.V.); 2LASIGE, Faculdade de Ciências Universidade de Lisboa, 1749-016 Lisboa, Portugal; sfalves@fc.ul.pt (S.A.); tjvg@di.fc.ul.pt (T.G.); lmcarrico@fc.ul.pt (L.C.); 3Unbreakable Idea Research, Lda, 2550-426 Painho, Portugal; 4Laboratório de Investigação em Microbiologia na Saúde Ambiental (EnviHealthMicro Lab), Instituto de Saúde Ambiental (ISAMB), Faculdade de Medicina, Universidade de Lisboa, 1649-028 Lisboa, Portugal; ccaneiras@gmail.com; 5Healthcare Department, Nippon Gases Portugal, 2600-242 Vila Franca de Xira, Portugal; 6Instituto de Medicina Molecular, Faculdade de Medicina, Universidade de Lisboa, 1649-028 Lisboa, Portugal; 7Neurology Service, Department of Neurosciences and Mental Health, Hospital de Santa Maria, Centro Hospitalar Universitário Lisboa Norte, 1649-028 Lisboa, Portugal

**Keywords:** virtual reality, cognition, transfer capacity, recovery of function, neurorehabilitation

## Abstract

Mild cognitive impairment (MCI) is characterized by cognitive, psychological, and functional impairments. Digital interventions typically focus on cognitive deficits, neglecting the difficulties that patients experience in instrumental activities of daily living (IADL). The global conjecture created by COVID-19 has highlighted the seminal importance of digital interventions for the provision of healthcare services. Here, we investigated the feasibility and rehabilitation potential of a new design approach for creating highly realistic interactive virtual environments for MCI patients’ neurorehabilitation. Through a participatory design protocol, a neurorehabilitation digital platform was developed using images captured from a Portuguese supermarket (NeuroVRehab.PT). NeuroVRehab.PT’s main features (e.g., medium-sized supermarket, the use of shopping lists) were established according to a shopping behavior questionnaire filled in by 110 older adults. Seven health professionals used the platform and assessed its rehabilitation potential, clinical applicability, and user experience. Interviews were conducted using the think-aloud method and semi-structured scripts, and four main themes were derived from an inductive semantic thematic analysis. Our findings support NeuroVRehab.PT as an ecologically valid instrument with clinical applicability in MCI neurorehabilitation. Our design approach, together with a comprehensive analysis of the patients’ past experiences with IADL, is a promising technique to develop effective digital interventions to promote real-world functioning.

## 1. Introduction

Mild cognitive impairment (MCI) was initially conceptualized as a clinical entity affecting the cognitive functioning exclusively (i.e., memory capacity) [1]. However, empirical evidence has shown that cognitive and functional impairments co-exist from very early stages of the disease [2], and difficulties in instrumental activities of daily living (IADL) are prevalent among MCI patients [3]. Vascular mild cognitive impairment (VaMCI) is a clinical condition of vascular etiology, in which executive deficits are a prominent feature [4,5] and a strong predictor of functional decline and dementia [6]. Nonetheless, studies conducted in VaMCI patients are scarce, with even fewer studies focused on the development of rehabilitation instruments which target cognitive and functional impairments simultaneously.

Virtual reality (i.e., computer-generated interactive environments) [7] has been recognized as a valuable resource for developing instruments that enable health professionals to accurately predict patients’ performance in everyday living activities (i.e., ecologically valid instruments) [8,9], especially in activities that actively engage executive functions [8,10,11,12]. Several virtual environments (VEs) have been developed targeting IADL, such as preparing meals [13,14], moving within the community [15,16], and cleaning and maintaining the house [17,18,19,20].

Shopping for groceries is perhaps one of the most studied IADL, with several studies showing that patients with executive deficits reveal in VEs a similar pattern of impairments to those observed in real-world tasks [21,22,23,24,25,26,27]. However, the majority of these VEs are manually designed, time- and human-resource consuming, and do not provide tasks or scenarios of sufficient realism [28]. Additionally, only a few of these VEs are designed for neurorehabilitation purposes [23] in MCI patients [22].

The feeling of “being physically present” in the virtual world, known as presence, is described as the phenomenon of users acting and experiencing emotions as if they were in the real world [29]; this is thought to promote the transfer of trained skills and behaviors from VEs to real-world contexts [8,29]. One design approach that has been used to improve the realism and sense of the presence is image-based rendering VEs. One example of a realistic image-based VE is Google Street View, where users can navigate within 360° photos of the surrounding environment. Previous findings have shown that the use of images of real-world scenarios results in highly visual realistic VEs and, therefore, an increased sense of presence [28]. Furthermore, image-based rendering VEs have shown promising results when applied to reminiscent therapy [28]. However, in the scarce studies that use this technique, the images are embedded as just a decorative/background element [30] or are non-interactive [15]. Other image-based VE interventions require complex technologic equipment and space availability in order to accommodate the experimental apparatus [28], which limits its widespread use in clinical and neurorehabilitation contexts.

Based on the limitations mentioned above and considering the concept of function-led instruments [8], in which neuropsychological instruments should be as far as possible an accurate representation of real-world functioning, we conducted a qualitative analysis of the rehabilitation potential and clinical applicability of an image-based fully navigable and interactive virtual supermarket, NeuroVRehab.PT to promote cognition and functional capacity in VaMCI patients. This neurorehabilitation platform was developed by ISAMB (Lisbon, Portugal), LASIGE (Lisbon, Portugal), and Nippon Gases Portugal (Vila Franca de Xira, Portugal).

The goals of this study are twofold: (1) to describe the design process of NeuroVRehab.PT and (2) to characterize health professionals’ (i.e., neurologists, psychologists, and neuropsychologists) perspectives about the rehabilitation potential and clinical applicability of NeuroVRehab.PT for VaMCI patients’ neurorehabilitation.

## 2. Experimental Section

This study was conducted using a participatory design research protocol and reported according to the consolidated criteria for reporting qualitative research checklist (COREQ) guidelines [31] (see Appendix A. COREQ: consolidated criteria for reporting qualitative research checklist). Two groups of stakeholders (i.e., older adults and health professionals) were invited to participate in the design of a virtual supermarket aiming to train the cognitive functions and behavior strategies recruited during a shopping activity. 

The present study was performed in compliance with the Declaration of Helsinki and was approved by the following ethics committees: Comissão de Ética do Centro Hospitalar de Lisboa Norte e Centro Académico de Medicina de Lisboa–CAML (reference number 89/19) and Comissão de Ética para Recolha e Proteção de Dados de Ciências (CERPDC) (reference number CERPDC/16/2019) (see Appendix A, ethical standards, for a description of the ethical aspects considered during the execution of this project).

### 2.1. Participatory Design of NeuroVRehab.PT–Shopping Behaviors Questionnaire with Older Adults (Phase 1)

#### 2.1.1. Sample and Recruitment

Two senior universities of the municipality of Almada, Portugal, were contacted and agreed to participate in the study. Visits were scheduled to a group of classes identified earlier by the executive board of the two institutions. During these visits, one of the co-authors of this paper (FFB) presented the project and explained how the data gathered through the questionnaires would support the research team’s decisions regarding the main features of the virtual supermarket, such as the use (or not) of a shopping list, the number of products included in the shopping list, and the type of supermarket (grocery store vs. supermarket vs. hypermarket), among other features. To be eligible, the participants had to be more than 60 years old, be community dwelling, be responsible for grocery shopping, and give written informed consent.

#### 2.1.2. Instruments and Procedure

A questionnaire with 11 multiple-choice questions that aimed to analyze the shopping behaviors and routines of older adults was developed and reviewed by the research team (see Appendix A for the shopping behaviors questionnaire with older adults). The questionnaire was filled in individually and collected at the end of the senior universities’ sessions. Questions included items regarding the type of store they usually go to (i.e., local grocery store, supermarket, or department store), the time spent there, and the frequency they go shopping per week. Other items included the habit of using shopping lists, establishing budgets, shopping for weekly or monthly necessities, and an estimation of the amount of money spent while shopping. The questionnaire was anonymous, and besides age, gender, and professional status (active vs. retired), no other personal information was collected.

#### 2.1.3. Data Analysis

Statistical Package for the Social Sciences (IBM-SPSS, version 24.0; International Business Machines Corp., Armonk, New York, NY, USA) was used to conduct descriptive analyses of the data collected through the shopping behaviors questionnaire with older adults. For nominal variables (e.g., gender, professional status, and questionnaire responses), tables of frequencies were calculated. For continuous variables (e.g., age, time spent shopping), the mean, standard deviation, mode, minimum, and maximum were calculated.

#### 2.1.4. Results

A total of 110 participants aged between 61 and 86 years (70.92 ± 5.94 years) filled in the questionnaire. Twenty-eight were men (26.7%) and 77 women (73.3%). Fifty-three participants (53.5%) stated that they usually go to supermarkets as opposed to local grocery (*n* = 8, 8.1%) stores or big department stores (*n* = 38, 38.4%). Fifty-nine (56.2%) claimed that they usually make shopping lists, but only 42 (75%) use it during shopping. Before going shopping, 44 (79.4%) participants claimed that they included items in the shopping list as they remember what they need, and 10 (14.7%) organized the products according to the products’ position in the supermarket.

More than half of the participants claimed they go to the supermarket less than once per week (*n* = 56, 52.8%), alone (*n* = 64, 60.4%), and to buy groceries for weekly necessities (*n* = 59, 56.2%). On average, the participants spend 59.24 ± 33.68 min on each visit to the supermarket. Sixty-one participants (58.1%) claimed that they have an estimation of how much they will spend before going to the checkout counter, and 33 (35.9%) stated that, after being informed of the value of the bill, they know precisely how much they should receive in return (see Appendix A for the shopping behaviors questionnaire in older adults descriptive data).

#### 2.1.5. Main Implications for the Development of the NeuroVRehab.PT

Supermarket was the most frequently visited type of store for food and house products shopping. A relevant percentage of the participants reported that they carry out their weekly shopping alone. The findings also suggested that shopping lists do not tend to be long, although diversified.

The majority of older adults in our sample claimed that they go to the supermarket less than once per week and spend around 60 min on each visit. These data were initially collected to determine an adequate dose exposure to our digital platform. Although there are no guidelines regarding dose exposure in cognitive rehabilitation for MCI patients [32,33], previous studies have shown that interventions composed of few sessions with extended durations were not effective [34]. This implies that clinical trials of cognitive rehabilitation should accommodate the difficulties that VaMCI patients experience in their real-life activities (related or not to cognitive decline), provide sufficient repetition, and manage fatigue and frustration throughout the intervention length [35].

### 2.2. Design and Implementation of the NeuroVRehab.PT (Phase 2)

Based on the shopping behavior survey with older adults, we developed a first prototype of a web-based application called NeuroVRehab.PT. NeuroVRehab.PT allows people suffering from VaMCI to perform typical shopping tasks, such as creating shopping lists, navigating in a supermarket, adding products to the shopping basket, or sticking to a budget. The application was developed using HTML version 5.2. [36], JavaScript version ECMAScript 2019 [37], CSS version 3 [38], and PHP version 7.4.9 [39]. The supermarket was constructed using the Photo Sphere Viewer library [40] and panoramic photographs of the interior of a typical Portuguese supermarket (captured using the Google Street View app for Android, Google LLC., Mountain View, CA, USA); see Appendix A, ethical standards, for a description of the ethical aspects considered during the image capture and editing).

The platform is optimized to run on tablets. Tablets provide direct object manipulation (i.e., the person directly interacts with the target object using the fingers) and require less hand–eye coordination [41]. Older adults seem to prefer tablets to more traditional setups such as computers [42]. Previous studies showed that tablets are easier for older adults to use [43], even when they experience technological divide to some extent [44] or already present mild [20] or severe cognitive compromise (i.e., dementia) [45].

#### 2.2.1. Platform Description

NeuroVRehab.PT is a prototype of a web-based digital neurorehabilitation platform composed of three independent game modes—supermarket, recipes, and shopping list. The supermarket is the central part of the platform and the key component of the three game modes.

##### Supermarket Environment Description

To replicate a real-world scenario, the environment of this system is composed of 49 360° panoramic photographs, together with typical supermarket noises at the background. The background noise can be turned off at any time. Users can only navigate in the supermarket by activating the full-screen mode or touching the start button (which also activates the full-screen mode). This way, users can have a clean screen to prevent distractions, as reported in previous studies [46]. *Maria*, a virtual shopping assistant, will guide users for the first time and regularly provide information about the game mechanics and behavioral shopping strategies that can be used in real life. Users experience the virtual supermarket from a first-person perspective (i.e., without any intermediating avatar) and can walk through the 19 supermarket sections (e.g., vegetables, fruits, bakery, dairy, frozen food) using the arrows displayed on the screen (Figure 1). Navigation arrows are placed in the screen, at the corridors, and in the exact position where the users want to walk to. If necessary, users may zoom in and out to take a closer look at a product or use the autorotate button to locate themselves in the environment. To select a product, users touch the product and a tag with the product’s information (name, photo, category, description, and price) is displayed on the right side of the screen, together with an “add to the basket” button that allows users to add products to the shopping basket (Figure 2a).

##### Supermarket Game Mode

The system provides fourteen game levels that ask users to go shopping with a predefined shopping list (Figure 2b). Levels have different difficulty levels—easy, medium, and hard—which differ regarding the number of products to buy, the distance between the products, the available time to complete the level, and the presence or absence of background noise. In more advanced levels, there is also a budget to be met. Furthermore, the platform is designed to enable health professionals to create custom levels. Users need to purchase all the items of the shopping list (Figure 2c) to complete the level. If the added product is listed on the shopping list, users will hear a sound of positive feedback and earn points. If users try to add a product that is not on the shopping list, a message together with a sound of negative feedback will appear, and users will keep the same points.

After selecting all the products on the shopping list, users are asked to go to the checkout counter zone and pay the groceries by choosing one of the payment methods (cash or credit card). At the end of each level, a three-star rating is attributed based on the users’ performance (time and distance walked).

##### Recipe Game Mode

The recipe game mode allows users to shop for the ingredients that are required to cook a traditional dish. Users may select among six traditional recipes—one soup, four main courses, and one dessert (Figure 3a). Each recipe is identified by a name and a photo of the dish. As an extra step of difficulty, after selecting one of the recipes the participants are asked to organize the ingredients under the correct category (e.g., apples under fruit) (Figure 3b). Correct and incorrect sorting is identified by turning the ingredients green or red, respectively. The participants need to correctly organize all the ingredients before having access to the virtual supermarket and purchasing the ingredients of the recipe.

##### Shopping List Game Mode

In this game mode, users can create personalized shopping lists. A list ID is identified by a name customized by users, and they may add products by writing the name of the product in the search field. As they write, a drop-down list appears with products matching the search query (Figure 3c). To select an item, they have to touch it. After finishing their shopping list, the participants are asked to organize the ingredients under the correct category (similar to the activity described in the recipe game mode section) before moving on to the virtual supermarket.

### 2.3. Software Evaluation and User Experience (Phase 3)

#### 2.3.1. Sample and Recruitment

A purposive sample of health professionals (i.e., psychologists, neuropsychologists, and neurologists) identified through professional networking was contacted and invited to participate in the study. Health professionals were identified based on (1) their clinical and scientific experience (≥5 years) with ageing, age-related neurological disorders, and cognitive decline; (2) their knowledge of the main theoretical models of human cognition and neurorehabilitation; (3) their familiarity with computerized cognitive training or rehabilitation programs. All the participants who were contacted (face-to-face) agreed to participate, and an individual session was booked at the participants’ convenience (local and date). All the participants signed the written informed consent.

#### 2.3.2. Instruments and Procedure

Interviews took place in quiet and private rooms and lasted approximately 60 min. The participants sat in front of a desk where a Huawei MediaPad T5 tablet (Android 8) (Huawei Technologies Co. Ltd, Shenzhen, China) was supported horizontally at 25° degrees (approximately) using a tablet stand. The platform was run in the Google Chrome application (version 79.0.3945; Google Inc., Menlo Park, CA, USA). A neuropsychologist with previous experience in qualitative research (FFB) conducted the interviews, and a second researcher member (SA, computer scientist and PhD student) was also present in five of the seven interviews and ensured that the sessions went on without any technical problems.

Each session was divided into two moments. In the first part of the interview, the participants were encouraged to express their opinions and thoughts verbally—the think aloud method (TA) [47]—while using the platform. The interviewer (FFB) demonstrated the TA method to participants while exploring the Gmail website. Then, the participants were encouraged to use it on the NeuroVRehab.PT platform. The participants were free to explore the platform; however, sessions were conducted so that all the participants visited the three game modes (i.e., supermarket, recipes, and shopping list) and played at least two game levels in the supermarket mode. The second part of the session consisted of a semi-structured interview focused on the participants’ experience (i.e., user experience; UX) and the perceived clinical applicability and rehabilitation potential of the platform. 

#### 2.3.3. Data Analysis

The participants’ demographic data were analyzed using IBM-SPSS (version 24.0; International Business Machines Corp., Armonk, New York, United States). Interviews were transcribed for content analysis, and a list of codes was developed on the basis of two interviews (data grounded theory). This initial code system was created independently by two researchers (FFB and SA) and then discussed and merged into a comprehensive list. The remaining interview transcripts were coded independently by two researchers (FFB and SA) using the previously developed list of codes and an interrater reliability index of Cohen’s *k* = 0.82 was obtained by calculating the mean of Cohen’s kappa indexes per interview (Cohen’s *k* ≥ 0.80). This kappa index was used as indicative of strong interrater reliability in healthcare research [48]. Blocks of 20 lines were used as the unity of analysis for semantic thematic analysis purposes [49]. When completing the analysis, the code system was reorganized into broad categories (major themes) and respective subcategories (minor themes).

## 3. Results

Seven health professionals aged between 29 and 67 (47.14 ± 13.08 years) and with 17 years of professional experience (range: 5–40 years) were interviewed. Two participants were male and 5 were female, with academic backgrounds in medicine/neurology (*n* = 2) or in neuropsychology (*n* = 5). 

Two participants rated themselves as being very confident, four as being confident, and one as a little confident in using technology or new technological devices (*n* = 2, *n* = 2, *n* = 3, respectively). On average, the participants spent 16 ± 7.99 h (range: 7–30 h) browsing content on the internet and 0.86 ± 1.86 h (range: 0–5 h) playing video games per week. All the participants were familiar with at least one computerized cognitive training or rehabilitation program (e.g., Cogweb^®^, Rehacom^®^), and three participants (42.9%) reported that they use brain training games or computerized cognitive training programs (CCTP) in their professional practice (e.g., Fitbrain, Neuronation).

Four major themes (and seven minor themes) emerged from the semantic thematic analysis: experience with NeuroVRehab.PT, rehabilitation potential, potential barriers, and opportunities (see Table 1 for the complete code system, including minor themes). Due to technical problems related to audio recording, the second part of interview 2 was lost (90% of the semi-structured interview). Therefore, only the information collected during the TA phase of interview 2 was included in data analysis. Data saturation was obtained at the 5th interview, with no new codes identified in the last two interviews.

### 3.1. Experience with NeuroVRehab.PT

#### 3.1.1. Hedonic Experience and Presence

Overall, the participants expressed a positive attitude towards NeuroVRehab.PT and reported having fun and enjoying the platform. Different features of the platform stood out and were considered by the participants as appealing and relevant from a clinical point of view (e.g., the icon used to identify the different game levels (Figure 2b), the label with the product’s characteristics (Figure 2a), the zoom-in functionality, and the availability of a shopping list that the participants can check (Figure 1)). However, it was the high realism and visual complexity (and auditory stimuli) of the VE that captured the participants’ attention. All the participants but one explicitly mentioned this aspect during the interviews (and some of them more than once).

Participant 1: “I was expecting something more, rudimentary, but not the case, I think it was… the products were clear and colors vivid…”Participant 2: “Well, it is a very appealing image of the supermarket. It makes you want to explore it, doesn’t it? It has beautiful fruits.”Participant 4: “Ah! How cute (…) it really looks like a supermarket (…), maybe this is really a supermarket (…) The products are real, not drawings [as in 3D-modulated scenarios], I think it is good.”Participant 5: “[While performing a task in NeuroVRehab.PT] Ok, hot chocolate. Hot chocolate, now I have to find…. this is really realistic. In fact, it is real, I did not have this expectation.”Participant 6: “It seems very realistic, hyper-realistic.” (line 51) “The locations, the type of products, yes, yes, I think is quite realistic.”Participant 7: “The environment sound is very good; it really puts you inside of a supermarket (…) we know where we are, perfectly.”

#### 3.1.2. Usability

The participants identified and played the three game modes without relevant difficulties. However, the navigation controls and the sense of orientation inside NeuroVRehab.PT were identified as two aspects that should be improved in order to provide a more smooth and pleasant experience. For instance, some participants misinterpreted the meaning of the navigation controls and interpreted them as mandatory actions to progress in the game.

Participant 2: “The arrow appears, I assume this means I need to follow the arrows.”Participant 1: “… the arrow then… means that I need to go back to the fruit section?”

Other participants considered the arrows as hints to the location of the next product in the shopping list.

Participant 5: “The arrow helps to orient, doesn’t it? I did not understand if the arrow gives you a hint or if … Does it give you a hint?”

Regarding the sense of being oriented within the supermarket, the participants felt that it “… could be useful, for example, before starting (the game levels) that the person visits the whole supermarket, a kind of orientation exercise, to learn (the supermarket) more or less.” (Participant 1). Still on this subject, another participant referred to the famous experiment of Willard S. Small [50] to explain the importance of having the opportunity to explore (learning) the environment before starting any specific task: “… a person is in that environment; it is like a mouse when it is put on a place, like (…) a maze, something like that, it (the mouse) will explore, the first thing it will do is to explore the surroundings to get a perception. We are a little bit like mice. The first thing a person wants to do is… in fact, surrounded by these fruits, is to see what is around and understand…” (Participant 2). In this regard, the participants suggested that the presence of signs with the names of the sections and/or a map of the supermarket would improve the sense of orientation as well as the learning of the VE.

### 3.2. Rehabilitation Potential

The health professionals considered NeuroVRehab.PT a useful and innovative instrument for cognitive stimulation that they would recommend to their MCI patients. 

Participant 5: “[regarding another virtual supermarket for IADL rehabilitation] (…) it was prehistoric when compared to this one.”Participant 2 “… a good alternative for cognitive stimulation.”

#### 3.2.1. Cognitive Stimulation

The participants considered NeuroVRehab.PT “… sufficiently appealing and demanding for MCI patients…” (Participant 5) and comprehensive in terms of the cognitive functions stimulated—namely, executive functions (working memory, planning, decision making), memory, attention, spatial orientation, and math abilities.

Participant 5 “… working memory, of course, always pumping in my head (…) Manifestly, this also trains orientation….”. […] From an attentional point of view, it is quite demanding”.

In addition, the choice for a shopping activity as the core activity of this digital intervention was considered as “…meeting the necessities of this population.” (Participant 1) and “… important, crucial for [patients] daily-life, if they do not have someone to do the shopping for them, they have to do it themselves…” (Participant 7).

The realism of the scenario was also identified as an asset that could be used to promote patients’ motivation to comply with long and emotionally demanding rehabilitation programs.

Participant 7: “[regarding other CCTP] I think there is not an effort or an intent to be similar to the person’s daily life. Here I notice that effort (…) and that could be more motivating for the person who is doing the training”.

#### 3.2.2. Transfer Capacity

NeuroVRehab.PT elicited a sense of presence in some participants; “… I already knew that the milk would be closer to where eggs were, it is similar to other supermarkets where I go, I knew that… even if some fruits are not displayed on the fruit exhibitor, they are right there, it is essentially like… my experience in other [real] supermarkets.” (Participant 1).

Participants were divided concerning NeuroVRehab.PT transfer capacity to patients’ daily life. Four participants considered that the activities proposed on NeuroVRehab.PT are “… more easily generalized than the paper-and-pencil exercises that we often do.” (Participant 5) and “Once the person succeeds in the game, I think…, I would say that it is easy to transfer to real life.” (Participant 6). Other participants considered that the transfer of the trained skills could not be assumed only based on the similarities between the virtual and real-world environments. Finally, a third group of participants claimed that the transfer of trained skills is more likely to occur if the activity/exercise is meaningful within the patient’s life context. 

Participant 2: “It has to make much sense [to the patient] to have any impact or transfer to real life. And even with this software, it is either something that meets what the previous life of the person was, and it has any meaning to him/her, or it ends up [just] being an interesting game…”.

Finally, it was also pointed out that, despite being very realistic, our virtual supermarket is still a controlled environment (e.g., being the only customer, the absence of entropy caused by the presence of other buyers, people covering the products) and different from the supermarket frequented by the patient.

### 3.3. Potential Barriers

Among the potential negative psychological side effects that MCI patients might experience associated with the use of our platform, the one most referred to by the participants was frustration (verbalized by six participants), followed by stress (*n* = 4), fatigue (*n* = 2), anxiety (*n* = 2), and irritation (*n* = 1). Feelings of frustration were primarily associated with the process of learning how the platform works. Nonetheless, the participants considered that potential negative psychological side effects could be easily managed if a therapist is present and supports the patient’s learning process by explaining game mechanics and controls and helping the patient cope with feelings of frustration.

Participant 7: “I believe they [the patients] will have some doubts (…) I had someone by my side to explain it to me, and they do not have it”.Participant 3: “I think in some cases, some patients can easily start to get frustrated. And it is advisable not to continue. And I think there will be some situations like that, which is perfectly normal with this type of activity. Therefore, it is necessary to have resources [as therapists] (…) to be you to finish the activity and help the patient to move on …”

Nausea was also referred to (*n* = 1) as a potential negative side effect as a result of the temporary blurred vision associated with the process of going from one image (sphere) to another. No other physical adverse effects (e.g., falls, dizziness) were foreseen by the participants, although difficulties associated with ageing, such as low visual acuity, were referred to.

### 3.4. Opportunities

Throughout the experimental sections, the participants identified three other possible applications of NeuroVRehab.PT.

#### 3.4.1. Clinical and Non-Clinical Contexts

NeuroVRehab.PT was considered as having applicability in other clinical populations, such as Traumatic Brain Injury (TBI), stroke, and early-stage dementia patients. Some participants also suggested that the platform could be used as a means to increase technology literacy in healthy older adults.

#### 3.4.2. Friend Sourcing

Three participants reported they would like to see some degree of interaction between users. One of the suggestions was to provide patients with the possibility of sharing recipes and cooking tips with other NeuroVRehab.PT users (i.e., patients), thus creating a virtual community. The participants also stated that it would be interesting if the therapist could create shopping lists (participant 5) that reflect the patient’s diet and/or food restrictions (participant 3). The fact that NeuroVRehab.PT is an online platform was identified by another participant as a way to promote family engagement, especially from younger people. “This (NeuroVRehab.PT) would be much more playful and (…) maybe it could even be more interesting to attract family participation (…) even for grandchildren, it would be much more interesting than sitting next to the grandmother, with a paper and pencil activity” (Participant 7).

#### 3.4.3. Other Environments

The possibility of exploring other IADL (e.g., finances, ATMs, housekeeping) as well as real-life scenarios (e.g., the supermarket where they usually go) was noted as being one of most significant opportunities that the design approach presented here could bring to the field of neurorehabilitation. Other proposed activities were to invite patients to cook the recipes available in the recipe game mode, explore public places, and train their capacity to use public transportation.

## 4. Discussion

The use of photo-realistic interactive virtual environments is a promising approach to develop accessible, cost-effective, and ecologically valid instruments for neurorehabilitation in VaMCI patients. The availability of digital instruments has gained even more prominence with the current global health conjecture due to the COVID-19 pandemic, where contact between health professionals and patients is restricted or drastically reduced. In this paper, we described the design and development process of a fully navigable and interactive virtual supermarket built from photos of a typical Portuguese supermarket—the NeuroVRehab.PT. Furthermore, health professionals with extensive clinical and research experience in neurodegenerative and age-related disorders assessed our platform and identified the advantages and challenges associated with its clinical use in VaMCI neurorehabilitation.

The participants considered NeuroVRehab.PT a remarkable improvement in the design of VEs for neurorehabilitation. The use of real-world supermarket photos as a core element of NeuroVRehab.PT resulted in a highly realistic and ecologically valid VE. In previous studies, the ecological validity of virtual supermarkets for assessment or training purposes was established through the relationships obtained between the participants’ performance in the VE and in related measures of executive functions and everyday functional capacity [23,26,51,52]. However, in the present study the ecological validity of NeuroVRehab.PT was established based on the verisimilitude approach [53]. According to this approach, ecological validity can be established based on the degree to which the demands of an experimental task resemble the cognitive demands of that task in the real world [53,54]. In this regard, not only is NeuroVRehab.PT an accurate representation of a typical Portuguese supermarket (face validity), but it was also claimed by the participants that our platform looked like (and felt like) a real supermarket. This was most evident in the statements of participants 1 and 5, when they said that their experience with NeuroVRehab.PT was very similar to shopping in a real-world supermarket; that is, they experienced the same difficulties and resorted to the same problem-solving strategies as they would in a real-world shopping activity. From this perspective, the design approach presented in this paper can also be applied to the development of instruments aiming to assess cognitive functioning, particularly executive functioning.

Moreover, our platform allowed the participants to interact and choose between grocery products that they are familiarized with and use in their daily life. This additional layer of customization (e.g., through the creation of personalized shopping lists that reflect real-life needs) enables coupling between therapeutic exercises and the patients’ interests and routines, thus resulting in meaningful exercises with practical (real-life) applications [35]. In other words, NeuroVRehab.PT is therefore a flexible rehabilitation platform to accommodate patients’ personal (e.g., food preferences), health (e.g., diet, food allergies and intolerances), cultural/religious (e.g., regional gastronomy), and socioeconomic status (e.g., branded vs. white label products).

Another layer of NeuroVRehab.PT aiming to promote patients’ engagement is the use of gamification processes. Gamification (i.e., the use of game elements in non-game contexts) [55,56] has been widely used as a means to promote patients’ adherence [57], self-esteem, satisfaction, and positive emotional experience [58] with healthcare interventions. In NeuroVRehab.PT, we implemented two types of game elements: numeric (e.g., points, scores) and visual (e.g., messages, three-star classification) feedback elements and narrative contexts. Although the study of the motivational influences of different game elements is still in its infancy, preliminary studies suggest that immediate feedback elements are associated with high adherence rates during the initial phases of contact with the platform [59,60]. On the other hand, the storyline in which the main action unfolds is thought to be a crucial feature for long-term adherence [60] (see reference [59] for a description of the rationale underlying the selection of this game element). Although the conclusions that can be drawn from a single moment of contact with our platform are limited, the participants’ statements that they had fun with and enjoyed using our platform is indicative that the implementation of these game-elements was successful, at least to some degree.

From the semantic thematic analysis, two subthemes emerged regarding the rehabilitation potential of our platform: cognitive stimulation and transfer capacity. Health professionals considered that the activities proposed in NeuroVRehab.PT are feasible by VaMCI patients and target the cognitive functions recruited during a shopping activity (e.g., orientation, planning, decision making, working memory, attention, math abilities). Furthermore, the participants referred to the training provided by our platform as being more easily generalized to the patient’s daily life when compared to other forms of cognitive stimulation, such as CCTP and Brain Training Games (BTG). Despite the undeniable social and economic impact of CCTP and BTG, the evidence regarding the efficacy of these programs is still scarce, with few studies showing an impact on other cognitive functions beyond those directly targeted by the cognitive training program, and even fewer studies showing an impact on behavioral outcomes [32,61,62,63].

It is noteworthy that the theoretical framework used in NeuroVRehab.PT highly contrasts with the one used in CCTP and BTG. While, in CCTP and BTG, individual exercises are developed to target isolated cognitive functions [64,65,66], in our platform we focused on developing a training exercise that aims to stimulate/train the adequate behavioral patterns/responses to perform one specific IADL successfully. Therefore, in NeuroVRehab.PT we privileged activity segmentation into its basic units (e.g., creating a shopping list, searching for products, checking the shopping) as a means to promote the patient’s awareness of where and how errors occur, along with the availability of simple strategies that patients could apply when shopping in the real world (e.g., organizing the shopping list and checking it before leaving each supermarket section). Promoting patients’ awareness, training practical strategies to overcome or avoid errors, and the periodical monitorization of action and goals are crucial steps in the Goal Management Training paradigm (GMT) [67]. The GMT has been used in the cognitive rehabilitation of patients with attention and executive deficits [67], including MCI patients, with positive results on quality of life [68], capacity to identify relevant occupational goals and efficient strategies, and the monitorization of task progression [69].

Nonetheless, not all the participants were sure about the transfer capacity of NeuroVRehab.PT. For instance, participant 3 stated that patients’ predisposing characteristics (e.g., previous shopping experiences) rather than the VE characteristics would influence patient adherence and skill transfer. The perspective that different people need and want to (re)learn different things and do this in different ways, using different strategies, is an established fact in the neurorehabilitation field, and programs should be committed to identifying which activities patients perceive as relevant in the actual context of their life [70]. In other words, no matter how well a system is designed or optimized to perform a task, it needs to relate to their users and correspond to their expectations [71]. Shopping for groceries is still a gender-based established activity, at least among older Portuguese generations. This will negatively influence how comfortable some older adults will feel performing a task that they are not used to, or that they might consider as conflicting with his/her role in the family and the social system. Therefore, in future feasibility and efficacy digital health intervention studies, factors such as perceived usefulness, expectations regarding digital interventions, and the perceived social impact of resorting to mental health programs [72] should be taken into account when establishing the participant inclusion criteria. 

Although NeuroVRehab.PT has been designed to be used independently by VaMCI patients, health professionals were in agreement regarding the importance of the presence of the therapist during the training sessions (at least, in the first sessions). The therapeutic bond established between the health professional and patient (i.e., therapeutic alliance) is a strong predictor of psychotherapy outcome [73], the decision to start treatment, and patient attrition [74,75,76]. Feelings of frustration triggered by successive experiences of failure are common among patients with cognitive impairments and, when not managed properly, can lead to feelings of confusion, anxiety, and dropouts [76]. In our study, the need for the therapist’s presence was associated with two main roles: to support the learning process of how the platform works, and to help patients cope with failure and frustration. This finding is supported by an extensive body of literature that shows that learning can be fostered when guided by a more experienced agent [77].

Furthermore, we acknowledge that VaMCI patients may also experience other difficulties observed in healthy older adults, such as the fear of failure and peer judgment, anxiety about using computers, and a low assessment of their skills and capacity to learn new things [78]. Therefore, an intermediate phase of user testing with healthy older adults is being planned to analyze age-related usability issues, gender and social variable influences, and the system stability [79], which may compromise the use of NeuroVRehab.PT by VaMCI patients.

## 5. Conclusions

Taken together, our results show that the NeuroVRehab.PT is an engaging, ecologically valid neurorehabilitation digital platform for VaMCI patient neurorehabilitation. Although some interface components such as the navigation controls should be optimized to improve the patients’ UX, health professionals considered this platform a significant step forward to the design of efficient and family-inclusive digital interventions for cognitive stimulation and IADL training. The present study also highlighted the impact that the perceived personal and social relevance of the training activities might have on patients’ adherence and long-term use and, ultimately, the interventions’ efficacy. Finally, our results are elucidative regarding the potential positive impact of the therapeutic bond between the health professional and patient on an intervention’s outcomes, and so this should not be dismissed even in interventions supported mostly by digital resources.

## Figures and Tables

**Figure 1 jcm-09-03821-f001:**
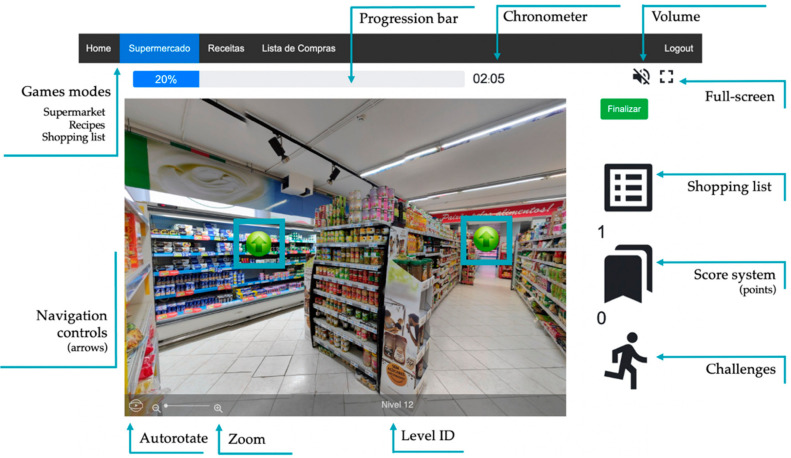
Interface elements of the NeuroVRehab.PT neurorehabilitation digital platform.

**Figure 2 jcm-09-03821-f002:**
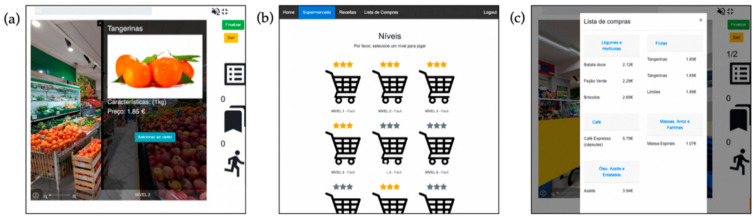
(**a**) Tag with product information—name, photo, description/weight, price, and add button; (**b**) view of the screen with the levels of difficulty, three-star classification, level ID, level of difficulty; (**c**) shopping list from a difficult level (medium) with nine products (written in black) under the correspondent category (written in blue).

**Figure 3 jcm-09-03821-f003:**
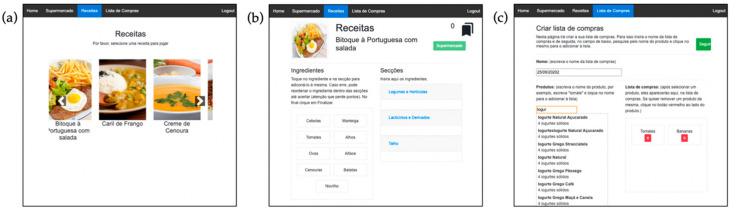
(**a**) View of the recipe game mode; (**b**) extra step of difficultly in the recipe game mode, in which users have to organize the ingredients (left side of the screen) under the corresponding category (right side of the screen); (**c**) view of the shopping list game mode with two products included in the list.

**Table 1 jcm-09-03821-t001:** Code system developed with the semantic analysis of the interviews.

⇒ Experience with NeuroVRehab.PT Hedonic experience and presence Usability ⇒ Rehabilitation potential Cognitive stimulation Transfer capacity ⇒ Potential barriers ⇒ Opportunities Clinical and non-clinical contexts Friend sourcing Other environments

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
