# Peer review of "Photo-Realistic Interactive Virtual Environments for Neurorehabilitation in Mild Cognitive Impairment (NeuroVRehab.PT): A Participatory Design and Proof-of-Concept Study"

_jcm, 2020, doi:10.3390/jcm9123821_

Round 1

Reviewer 1 Report

  1. “Results section, line number 272”. Kindly correct the values of ‘n’. I think it should be n=2, n=4, n=1.
  2. “Results section, line number 276-277”. It has been mentioned that:

‘they use brain training games or computerized cognitive training programs (CCTP) in their professional practice.’ Kindly specify which brain training games or CCTP? Any specific name?

  1. “Results section, line number 278”. It has been mentioned that:

‘Four major themes (and seven minor themes) emerged from the semantic thematic analysis.’ Kindly elaborate. What are major and minor themes? Enlighten more about semantic thematic analysis.

  1. Throughout the manuscript, the participants’ sayings are followed by line numbers in the bracket. What does that line number mean? For instance, refer to lines 295-299.

‘Participant 2: “Well, it is a very appealing image of the supermarket. It makes you want to explore it, doesn´t it? It has beautiful fruits.” (line 24)

Participant 4: “Ah! How cute (…) it really looks like a supermarket (…), maybe this is really a supermarket (…) The products are real, not drawings [as in 3D-modulated scenarios], I think it is good.” (line 52)’

In the above-mentioned examples what does (line 24) and (line 52) show?

  1. “Heading 3.2.1. Cognitive Stimulation”. How cognitive stimulation is measured?
  2. “Heading 3.2.2. Transfer Capacity”. How it comes under ‘Rehabilitation Potential (heading 3.2)’. What is the link between Transfer Capacity and Rehabilitation Potential?
  3. “Heading 3.2.2. Transfer Capacity, line 358”. It has been mentioned that:

‘Some participants considered that the activities proposed on NeuroVRehab.PT is’. Don’t use the word ‘Some’. Be specific about the number of participants.

  1. “Heading 3.3. Potential Barriers”. Why do the participants feel stress, fatigue, anxiety, and irritation? Explain the reasons for the following health conditions and also describe that these medical conditions are observed against which game mode?
  2. In the presented research, a NeuroVRehab.PT is developed based on the shopping behavior survey with older adults but it is tested only on a small sample of health professionals. Why testing is made only on health professionals and not on other older adults? How the overall credibility of the developed platform will be justified if only health professionals (with small sample size) are considered for testing?
  3. In the result section of the manuscript (heading 3, starting from line 266), a generalized result has been reported throughout the section. Kindly categorized the results based on game modes as well. In every result, also mention which game mode has been used. Make sure that the comparison between participants’ responses should be made only for the same game mode.
  4. “Heading 3.4.1. Clinical and non-clinical contexts, line 392-393”. It has been mentioned that:

‘NeuroVRehab.pt was considered as having applicability in other clinical populations, such as Traumatic Brain Injury (TBI), Stroke, and early-stage Dementia.’ How the developed system could be beneficial for TBI, stroke, and dementia patients? Provide an elaborative explanation.

  1. “Heading 3.4.2. Friend Sourcing, line 396”. Don’t use the word ‘Some’. Be specific about the number of participants.
  2. “Heading 5. Conclusion, line 517-518”. It has been mentioned that:

‘Taken together, our results show that NeuroVRehab.PT is an engaging, ecologically valid neurorehabilitation digital platform for VaMCI patients neurorehabilitation.’ Without application testing on VaMCI patients, how the authors claim that it is a valid neurorehabilitation digital platform for VaMCI patients? The patients’ opinions could be different from that of health professionals.

  1. What is the impact of gender on reported results? As only 2 males were involved in the testing phase, so how can you differentiate between the male and female behavior while using the platform? Kindly provide a detailed explanation.
  2. Which game mode is most and least effective for neurorehabilitation? Present the experimental results.
  3. For the credibility, feasibility, and acceptance of results, the testing of the developed platform on VaMCI patients is necessary.
  4. The picture quality throughout the manuscript is poor. Kindly improve the picture quality (for instance, Figure 2 and 3)
  5. Overall language of the manuscript should be improved

Reviewer 2 Report

Overall, I really enjoyed this paper.

I think studies like this are extremely important to consider when designing and exploring rehabilitative interventions. I enjoyed learning about your process of developing the NeuroVRehab.PT program, and all the careful thought and considerations that your team went through to make it an effective intervention. 

The authors did a good job explaining their process in designing a tablet based cognitive ADL rehabilitation program. My one main criticism is that while I completely agree that this program would be beneficial for individuals with mild cognitive impairment (MCI), I was confused while the authors choose to specify vascular MCI as their target population and in their title, when no clinical population was involved in the study. However, I'm confident with minor to moderate revisions the authors will address this concern. 

Reviewer 3 Report

No.

Reviewer comment

1

Good and interesting paper

2

Introduction:

It should be explored a little more: 1) why you refer only to VaMCI and not in general to MCI; 2) what does it with ‘virtual rehabilitation’ and the concept of ecologically instrument

3

Introduction:

At line 48 you say ‘low quality of life’; but if you are talking about MCI you usually don’t have ‘low quality of life’; could you explain that, please?

4

Line 390

3.4. Opportunities

There is no words in this section: why?

5

You should consider the possibility to administer NeuroVRehab.PT by modulating different levels of difficulty and complexity both as regards the levels of usability (think of older people and less like to navigate in a virtual reality environment) and as regards the different levels of cognitive impairment

6

3.2.1. Cognitive Stimulation

Participants considered NeuroVRehab.PT “… sufficiently appealing and demanding for MCI 336 patients…” (Participant 5, line 547) and comprehensive in terms of the cognitive functions stimulated namely: executive functions (working memory, planning, decision making), memory, attention, 338 spatial orientation, and math abilities.

The authors talk about cognitive functions involved. Are there any performance indicators in your tool? so as a professional could be able to evaluate the progress of the treatment and modulate any dimensions? if you want to carry out a pilot test on VaMCI subjects, you could administrate a cognitive assessment with paper-and-pencil tests that evaluate these aspects and then do some analysis with some performance index.
